# Suicide fatalities in the US compared to Canada: Potential suicides averted with lower firearm ownership in the US

**Julia Raifman[1]\*, Laura Sampson[2], Sandro Galea[3]**

**1** Department of Health Law, Policy, and Management, Boston University School of Public Health, Boston, Massachusetts, United States of America, **2** Department of Epidemiology, Boston University School of Public Health, Boston, Massachusetts, United States of America, **3** Boston University School of Public Health, Boston, Massachusetts, United States of America

\* jraifman@bu.edu

**Data Availability Statement:** All relevant data are within the paper and its Supporting Information files.

**Funding:** The author(s) received no specific funding for this work.

## Abstract

### Introduction and objective

The United States (US) has the highest rate of firearm suicides in the world. The US and Canada are comparable countries with markedly different rates of firearm ownership, providing an opportunity to estimate suicide fatalities that could be averted in the US with a lower rate of firearm ownership.

### Methods

We compared 2016 US suicide fatality rates–standardized within fourteen sex-specific age groups to reflect the ethnic composition of Canada–to 2016 Canadian suicide rates. We then calculated the number and proportion of suicides that could be averted in the US if the US had the same rates of suicide as in Canada.

### Results

If the US had the same suicide rates as in Canada, we estimate there would be approximately 25.9% fewer US suicide fatalities, equivalent to 11,630 suicide fatalities averted each year. This decline would be driven by a 79.3% lower rate of firearm-specific suicide fatalities. The male suicide fatality rate would be 28.8% lower and equivalent to 9,992 fewer suicide fatalities each year. The female suicide fatality rate would be 16.0% lower and equivalent to 1,638 fewer suicide fatalities each year. While 36% of firearm suicide fatalities could be replaced by non-firearm suicide fatalities, 64% of firearm fatalities could be averted entirely.

### Conclusions

US policymakers may wish to consider policies that would reduce rates of firearm ownership, given that that about 26% of US suicide fatalities might be averted if the US had the same suicide rates as in Canada, a country with drastically lower firearm ownership rates.

**Competing interests:** The authors have declared that no competing interests exist.

## Introduction

Nearly 45,000 people died due to suicide in the United States (US) in 2016 [1], a rate that increased in 2017 and contributed to a general decline in US life expectancy [2]. Of US suicides, 51% were due to firearms. The US has the highest rate of firearm suicides in the world, with 35% of global firearm suicide fatalities but among just 4% of the world's population [3].

With an estimated 120.5 firearms per 100 people, US civilian firearm ownership per person is greater than that of any other country by more than twofold and accounts for 46% of global civilian firearm ownership [4]. Firearms are a particularly lethal means of suicide [5], and firearm ownership is linked to likelihood of suicide fatalities. In one US study, each one percent increase in state household firearm ownership over time was associated with 0.22 more suicide fatalities per 100,000 adults [6]. In another US study, between 1981 and 2002, each 10% reduction in regional household firearm ownership over time was associated with a 2.5% reduction in suicide fatalities [7]. These data, coupled with the unusually high rate of firearm suicides in the US, suggest that the high prevalence of firearms in the US may drive the high rate of suicide fatalities.

Removal of a particularly lethal means of suicide has reduced suicide rates in other settings. In the United Kingdom, breathing coal gas containing toxic carbon monoxide from ovens was the leading method of suicide until the 1960s, when the gas supply transitioned to natural gas. Eliminating coal gas as a readily accessible means of suicide was associated with reductions in coal gas related suicide fatalities of 80% among males and 87% among females, and reductions in overall suicide fatalities of 34% among males and 32% among females [8]. Similarly, bans of a highly fatal herbicide commonly used as a method of suicide in South Korea were associated with a 46% reduction in suicides due to herbicides or fungicides and a 10% reduction in overall suicide fatalities [9]. Similar reductions in suicide were observed with pesticide bans in Sri Lanka [10]. While there was some replacement of restricted means with alternate means of suicide in these areas, means restriction of common methods of suicide fatalities was linked to fewer overall suicide fatalities in each setting.

Other studies suggest there have been similar reductions in all-cause suicide fatalities after countries have changed federal policies to reduce firearm ownership or access. After Israel restricted weekend access to firearms for Israeli Defense Force soldiers, the suicide rate declined by 40% [11]. After Australia banned long guns and implemented a gun buyback program beginning in 1996, trends in all-cause suicide mortality reversed from increasing 1% percent per year to declining by 1.5% per year [12]. Finally, after Austria made handgun purchase policies more stringent in 1998, there was a decline in the suicide rate due to firearms, but not in the all-cause suicide rate [13].

These examples raise an important question: would the US witness fewer suicide fatalities if fewer firearms were available?

Evaluating how lower firearm ownership would affect US suicide fatalities is not straightforward. To do so, we need to estimate suicide fatality rates in a context with different levels of firearm ownership. While there is variation in firearm ownership across US states, those states that are most educated, wealthy, and likely to vote democratic tend to have the most firearm regulation and the fewest firearms [14]. The observable differences between states means there are likely unmeasured ways in which the states differ as well, making it difficult to determine whether lower firearm ownership or differences in other characteristics drive lower suicide fatality rates.

A better approach to answering this question is afforded by comparing the US to Canada. Canada is comparable to the US in demographic and macroeconomic characteristics, and is commonly used as a point of comparison for the US [15–18]. The US and Canada are both

democracies and among the top 20 countries in the world with regard to gross domestic product (GDP) per capita [19]. Eighty-nine percent of the US population and 87% of the Canadian population have a high school diploma, and 21% of the US population and 17% of the Canadian population have a university degree [20,21]. Fifty-three percent of the US population and 46% of the Canadian population are married [22,23]. Approximately 82% of the US population and 81% of the Canadian population live in urban areas; and the US unemployment rate is 4.9%, while the Canadian unemployment rate is 7.0%.

While similar in these respects, the US and Canada diverge in two important respects relevant to the question at hand.

First, the two countries differ markedly in firearm ownership. Federal firearm restrictions have made Canadian firearm ownership much lower than in the US across localities with varying incomes, educational attainment, and political views. The overall estimated firearm ownership rate is 120.5 firearms per 100 people in the US [4], in contrast to 34.7 firearms per 100 people in Canada. Canada restricts widespread access to firearms principally by requiring individuals to have a license to possess a firearm, a process that takes up to 45 days [24].

Second, key demographic groups that have distinct patterns of suicide–Black and Native populations–make up different proportions of the US and Canadian populations. Black populations typically have lower suicide fatality rates in the US [25], and a greater proportion of the US compared to the Canadian population is Black. Native populations have elevated suicide fatality rates [26], and a lesser proportion of the US population is Native relative to the Canadian population.

To estimate US suicide fatalities in a context of fewer firearms, we compared all-cause and firearm-specific US suicide fatality rates to Canadian suicide fatality rates while accounting for demographic differences, standardizing suicide fatality data in the two countries by ethnicity.

## Materials and methods

### Data and variables

Statistics Canada [27] provides data on the age group, sex, and method of suicide of individuals who die by suicide, but does not provide information on ethnicity of decedents within these demographic groups. The US provides more detailed data, including the age, sex, and ethnicity of those who die by suicide, along with the cause of suicide (firearm vs. non-firearm). To standardize the US data for comparison to Canada, we estimated the ethnic composition of the Canadian population and adjusted US suicide data to represent fatality rates that would be observed if the US population had the same ethnic composition as the Canadian population (henceforth referred to as the "US standardized population), as detailed below.

We obtained data on Canadian suicide fatalities from Statistics Canada through the Government of Canada Open Data website. We extracted age group- and sex-specific 2016 fatality data on firearm and non-firearm suicide from the "Deaths, by cause, Chapter XX: External causes of morbidity and mortality (V01 to Y89)" dataset [28]. We considered suicide fatalities with the following codes to be firearm-specific suicide fatalities: Intentional self-harm by handgun discharge; intentional self-harm by rifle, shotgun and larger firearm; and intentional self-harm by other and unspecified firearm discharge. We calculated non-firearm suicide fatalities by subtracting firearm-specific suicide fatalities from the total fatalities due to "intentional self-harm," within each age and sex group. We obtained data on the overall population in each sex-specific age group from the 2016 Canadian Census through Statistics Canada [29].

We obtained data to approximate the ethnic composition of the Canadian population by sex-specific age groups from two tables: Table (A) listed single and multiple ethnicity responses grouped by categories of Aboriginal (First Nations, Métis, and Inuit) vs. non-

Aboriginal origins [30] and Table (B) listed single and multiple ethnicity responses by reported continent and country of origin [31]. These tables did not provide information on the specific combination of ethnicities of individuals who reported multiple ethnicities, so we created three mutually exclusive categories. Within each sex-specific age group, we first calculated the number of individuals reporting any Aboriginal ancestries by adding Table (A) data on reports of "Aboriginal ancestry (only)" and "Aboriginal and non-Aboriginal ancestries." We then calculated the number of Canadians in each sex-specific age group who reported "African origins" based on the Table (B) data. Finally, we estimated the number of non-Aboriginal, non-African individuals by subtracting the African and Aboriginal numbers from the total number of individuals in each sex-specific age group. We determined the age groups for the full analysis based on the age groups for which Statistics Canada provided data on ethnicity. We used these data to approximately align Canadian data with three ethnicity groups in the US data: Native American, Black, and White or Asian.

We obtained data on US firearm and non-firearm suicide counts, population denominators, and crude suicide fatality rates by age group, sex, and ethnicity from the Web-based Injury Statistics Query and Reporting System (WISQARS) through the Centers for Disease Control and Prevention (CDC) [1]. WISQARS reported that there were three suicide fatalities among individuals of unknown age group, which were excluded from the analysis. To standardize death rates using Canadian ethnic distributions within each sex-specific age group, we approximated Canadian ethnicity groups using US ethnicity data provided in WISQARS. For both firearm and non-firearm suicide fatalities in each sex-specific age group, we aligned the "Black" category in the US data with the "African origins" category in the Canadian data. We aligned the "American Indian/Alaska Native" data from the US with the "Aboriginal" category in the Canadian data. Finally, we aligned the "White" and "Asian or Pacific Islander" groups in the US data to approximate the "non-Aboriginal, non-African" group from the Canadian data.

## Statistical analysis

For each sex-specific age group in the US, we created a standardized US population matching the ethnic composition of the Canadian population. To estimate suicide fatality rates in the standardized US population, we summed US ethnicity-specific estimates of firearm and non-firearm suicide fatalities, multiplied by the proportion of the Canadian population of a given ethnicity. We then multiplied the estimated rates by 100,000 to provide the suicide rate per 100,000 individuals. Each step of the standardization is described in additional detail in the (S1 Text and S1–S7 Tables).

We calculated all-cause and cause-specific suicide fatality rates per 100,000 people in Canada by dividing the number of suicide fatalities in each sex-specific age group by the population in each group and multiplying this rate by 100,000. We created a bar graph depicting the US standardized suicide fatality rates and Canadian suicide fatality rates for each sex-specific age group. We calculated 95% confidence intervals for the total suicide fatality rates using the normal approximation for the Poisson distribution [32,33], as is standard for rates, depicted as error bars in the figure.

For each sex-specific age group, we calculated rate differences by subtracting the Canadian suicide rates from the US standardized suicide rates for all suicides and firearm-specific and non-firearm-specific suicides. We then divided this difference by the US standardized suicide fatality rate to estimate the proportion of suicide fatalities that would be averted if the US had the same suicide rate as in Canada, for each sex-specific age group. Finally, we multiplied this proportion by the actual number of US suicide fatalities in each sex-specific age group, to estimate the fatalities that could be averted. We summed the fatalities averted in each sex-specific

age group to estimate the total potential fatalities averted by sex and in total for overall and cause-specific suicide fatalities, calculating additional fatalities rather than fatalities averted for non-firearm suicides. This analysis is described in detail with an example in the (S1 Text and S1–S7 Tables). We also estimated the proportion of firearm suicide fatalities for which there would be means replacement by subtracting the total fatalities averted from the firearm fatalities averted and dividing by firearm fatalities averted.

## Results

Table 1 shows the characteristics of the US and Canadian population in 2016. There were 35,151,740 individuals included in the Canadian Census vital statistics data (used for denominators for suicide rates); 34,460,060 individuals included in ethnicity tables from the Canadian Census (used for weighting); and 323,127,513 individuals in the US WISQARS data in 2016. Approximately 91% of Canadian males and females and 85% of US males and 84% of US females were non-Aboriginal and non-African, or White and Asian. Approximately 14% of US males and females and three percent of Canadian males and females were Black or of African origins, while six percent of Canadian males and females and 1.4–1.5% of US males and females were Aboriginal or American Indian/Alaska Native.

Fig 1 shows the suicide fatality rate per 100,000 people by age group in 2016 in Canada relative to the US population standardized to the ethnic characteristics of Canada, with part (a) depicting rates for males and part (b) depicting rates for females.

Relative to Canada, the male suicide fatality rate in the US standardized population was greater in every age group. The proportion of male suicide fatalities due to firearms in the US standardized population ranged from 46% among those aged 25 to 34 to 78% among those aged 65 or older. In contrast, the proportion of male suicide fatalities due to firearms in Canada ranged from six percent among those under the age of 15 years to 32% among those aged 65 or older. We estimated that if males in the US had the same firearm suicide fatality rate as Canada, we would observe 76.9% fewer suicide fatalities due to firearms and 32.8% more non-firearm suicide fatalities. The overall male suicide fatality rate would be 28.8% lower if males in the US had the same firearm suicide fatality rate as Canada, and would be equivalent to approximately 9,992 fewer suicide fatalities each year.

The female suicide fatality rate in the US standardized population was lower than that of Canada for those under age 25 and greater for those aged 35 or older. The proportion of female suicide fatalities due to firearms in the US standardized population ranged from 19% among those under age 15 to 36% among those older than 65 years. In contrast, the proportion of female suicide fatalities due to firearms in Canada ranged from less than one percent among those under the age of 15 years to 6% among those aged 55 to 64 years. We estimated that if females in the US had the same firearm suicide fatality rate as Canada, we would observe 93.4% fewer suicide fatalities due to firearms and 19.5% more non-firearm suicide fatalities. The female suicide fatality rate would be 16.0% lower if females in the US had the same firearm suicide fatality rate as Canada, which would be equivalent to approximately 1,638 fewer suicide fatalities each year.

We estimated that the suicide fatality rate would be 25.9% lower if the US had the same suicide fatality rate as Canada, driven by a 79.3% lower rate of firearm-specific suicide fatalities. The lower suicide fatality rate would be equivalent to 11,630 fewer suicide fatalities each year. There was evidence of means replacement for some but not most suicide fatalities, with a 36.0% of firearm suicide fatalities replaced by non-firearm suicide fatalities but 64.0% of firearm fatalities averted entirely.

**Table 1. Population characteristics in the United States and Canada, 2016.**

| Age Group | Sex | Ethnicity | Canada | | United States | |
|---|---|---|---|---|---|---|
| | | | N | Percent | N | Percent |
| 0 to 14 | Male | Aboriginal | 287,470 | 9.6 | 585,881 | 1.9 |
| | | African origins | 161,155 | 5.4 | 5,196,385 | 16.7 |
| | | Non-aboriginal, non-African | 2,532,520 | 85.0 | 25,353,242 | 81.4 |
| | Female | Aboriginal | 276,295 | 9.7 | 570,162 | 1.9 |
| | | African origins | 154,980 | 5.5 | 50,27,401 | 16.8 |
| | | Non-aboriginal, non-African | 2,404,625 | 84.8 | 24,241,998 | 81.2 |
| 15 to 24 | Male | Aboriginal | 172,120 | 7.9 | 389,861 | 1.7 |
| | | African origins | 84,150 | 3.9 | 36,88,506 | 16.5 |
| | | Non-aboriginal, non-African | 1,914,130 | 88.2 | 18,214,544 | 81.7 |
| | Female | Aboriginal | 172,400 | 8.4 | 374,352 | 1.8 |
| | | African origins | 81,380 | 3.9 | 3,560,736 | 16.8 |
| | | Non-aboriginal, non-African | 1,807,545 | 87.7 | 17,283,028 | 81.5 |
| 25 to 34 | Male | Aboriginal | 142,035 | 6.3 | 374,596 | 1.7 |
| | | African origins | 79,000 | 3.5 | 3,338,882 | 14.8 |
| | | Non-aboriginal, non-African | 2,043,930 | 90.2 | 18,886,260 | 83.6 |
| | Female | Aboriginal | 166,515 | 7.2 | 346,167 | 1.6 |
| | | African origins | 89,020 | 3.9 | 3,466,922 | 15.7 |
| | | Non-aboriginal, non-African | 2,056,075 | 88.9 | 18,264,416 | 82.7 |
| 35 to 44 | Male | Aboriginal | 125,615 | 5.7 | 309,569 | 1.5 |
| | | African origins | 85,310 | 3.9 | 2,702,577 | 13.4 |
| | | Non-aboriginal, non-African | 1,984,145 | 90.4 | 17,140,617 | 85.1 |
| | Female | Aboriginal | 148,225 | 6.4 | 297,098 | 1.5 |
| | | African origins | 87,705 | 3.8 | 3,017,848 | 14.9 |
| | | Non-aboriginal, non-African | 2,076,770 | 89.8 | 17,002,447 | 83.7 |
| 45 to 54 | Male | Aboriginal | 128,220 | 5.2 | 277,098 | 1.3 |
| | | African origins | 68,065 | 2.8 | 2,666,665 | 12.6 |
| | | Non-aboriginal, non-African | 2,246,375 | 92.0 | 18,162,376 | 86.1 |
| | Female | Aboriginal | 145,745 | 5.7 | 277,646 | 1.3 |
| | | African origins | 60,115 | 2.4 | 3,018,481 | 13.9 |
| | | Non-aboriginal, non-African | 2,343,455 | 91.9 | 18,384,413 | 84.8 |
| 55 to 64 | Male | Aboriginal | 101,825 | 4.3 | 216,738 | 1.1 |
| | | African origins | 36,965 | 1.6 | 2,284,809 | 11.4 |
| | | Non-aboriginal, non-African | 2,227,070 | 94.1 | 17,497,491 | 87.5 |
| | Female | Aboriginal | 114,835 | 4.6 | 235,033 | 1.1 |
| | | African origins | 30,300 | 1.2 | 2,712,845 | 12.6 |
| | | Non-aboriginal, non-African | 2,344,065 | 94.2 | 18,516,228 | 86.3 |
| 65+ | Male | Aboriginal | 69,390 | 2.7 | 169,296 | 0.8 |
| | | African origins | 25,160 | 1.0 | 1,866,064 | 8.6 |
| | | Non-aboriginal, non-African | 2,56,930 | 96.3 | 19,757,466 | 90.7 |
| | Female | Aboriginal | 79,835 | 2.7 | 206,658 | 0.8 |
| | | African origins | 24,625 | 0.8 | 2,758,899 | 10.1 |
| | | Non-aboriginal, non-African | 2,823,970 | 96.4 | 24,485,812 | 89.2 |
| Total | Male | Aboriginal | 1,026,675 | 6.0 | 2,323,039 | 1.5 |
| | | African origins | 539,805 | 3.2 | 21,743,888 | 13.7 |
| | | Non-aboriginal, non-African | 15,405,100 | 90.8 | 135,011,996 | 84.9 |
| | Female | Aboriginal | 1,103,850 | 6.3 | 2,307,116 | 1.4 |
| | | African origins | 528,125 | 3.0 | 23,563,132 | 14.4 |
| | | Non-aboriginal, non-African | 15,856,505 | 90.7 | 138,178,342 | 84.2 |

(*Continued*)

**Table 1.** (Continued)

| Age Group | Sex | Ethnicity | Canada | | United States | |
|---|---|---|---|---|---|---|
| | | | **N** | **Percent** | **N** | **Percent** |
| **Total** | | | **34,460,060** | | **323,127,513** | |

We used the Canadian descriptions of ethnicity in the table. "Aboriginal" corresponded to "American Indian/Alaska Native;" "African" corresponded to "Black;" and "Non-aboriginal, non-African" corresponded to "White" and "Asian" in the US data. The Canadian census only inquired about the ethnicity of individuals living in private residences, so the total is lower than the total population.

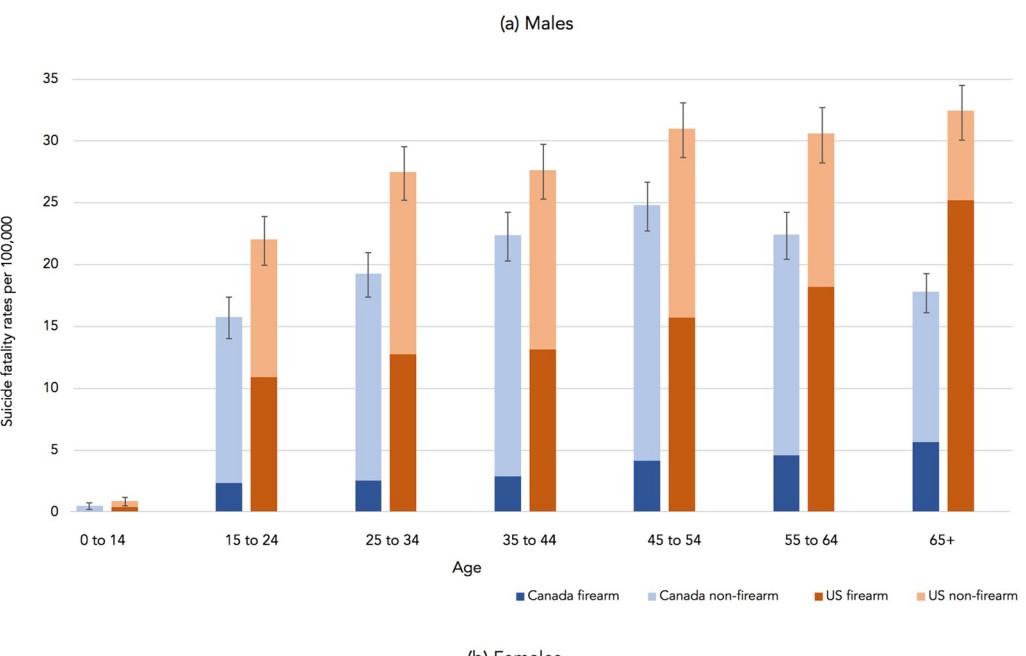

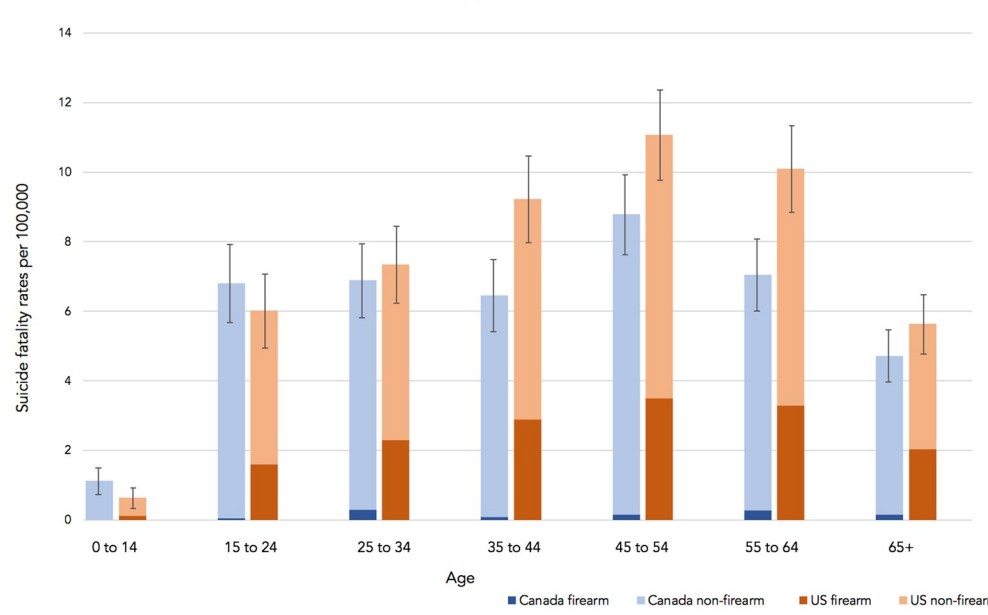

**Fig 1. Cause-specific suicide fatality rates in the US and Canada, by sex and age group.** Notes: The US population is the US standardized population, reflecting the demographic characteristics of the Canadian population.

## Discussion

We estimated that the US suicide fatality rate would be about 26% lower than it is now, with 11,630 fewer suicide fatalities each year, if the US had the same levels of civilian firearm ownership as Canada. A 79% reduction in firearm-specific suicide fatalities would drive the estimated difference in overall suicide fatality rates. The greater reduction in suicide fatalities among males (29%) compared to females (16%) is consistent with firearms accounting for a greater proportion of suicide fatalities among males than females. While an estimated 36% of firearm suicides in both sexes would be replaced with non-firearm suicides, most firearm suicide fatalities would be averted and not result in fatalities.

The estimated 26% reduction in suicide fatalities that may be achieved with firearm means restriction is plausible and consistent with the more than 30% reduction in suicide fatalities with coal gas means restriction in the United Kingdom [8] and reductions in suicide fatalities with pesticide means restriction in South Korea [9] and Sri Lanka [10]. The findings that means-restricted suicide fatalities would be partially replaced with suicide fatalities due to other means, but that there would be overall declines in suicide fatalities, is also consistent with the patterns of suicide fatalities documented after means restriction in the United Kingdom, South Korea, and Sri Lanka [8–10]. The reduction in suicides associated with reduced firearms is also consistent with evidence of reduced all-cause suicides following federal policy changes in Israel and Australia [11,12].

While we adjusted for differences between the US and Canadian population to the extent possible given the study design and availability of data, this study had many limitations. By comparing only two countries, we did not have the ability to control for differences between the two countries including poverty, unemployment, health systems, cultural, or other differences. To the extent that the US and Canada differ in some of these respects, Canada has lower GDP per capita, greater unemployment, lower educational attainment, and lower marriage, each of which are associated with elevated rates of suicide fatalities [34], suggesting that our results may be an underestimate of US suicide fatalities that would be averted in the context of Canadian firearm ownership. While we adjusted for sex, age group, and ethnicity, we could only control for the characteristics available in the Canadian suicide fatality and demographic data. Given that detailed suicide fatality data was not available by ethnicity in Canada, it is not clear whether the associations between ethnicity and suicide observed in the US (i.e., that white individuals are much more likely than black individuals to commit suicide) also hold in Canada. In other words, there may be effect modification by country in this relationship, which could affect our overall results and conclusions.

A further limitation was that the Canadian ethnicity data were collected only from those living in private residences and not collective dwellings, meaning the proportions may not be representative of the entire population of Canada. We were limited to data on three ethnic groups based on the overlap in ethnic groups reported in both the US and Canada, and potential misalignment between US data on "Black" and "American Indian/Alaska Native" individuals with Canadian data on individuals who reported having any African origins or being "Aboriginal." Those who reported "African origins" are unlikely to be the only individuals who are black in Canada, and some individuals who reported "African origins" may not be black. We also did not have data on whether individuals were both of "African origins" and "Aboriginal," so our approach assumes that no individuals were in both categories. Individuals with multiple ethnicities were only reflected in one ethnicity category in this analysis.

Finally, the US does not have data on circulating firearms or firearm certificates, and ultimately we cannot measure the exact effect of a change in firearm availability or ownership on suicide deaths. Our conclusions rest on the assumption that if the US had similar firearm

ownership rates as in Canada, the firearm suicide death rates in the US would be comparable to those in Canada, given the many other similarities between the two countries.

## Conclusion

The US and Canada are similar in many respects, with a notable exception being that Canada has markedly lower firearm ownership across settings, a difference that we drew on to estimate the proportion of suicide fatalities that might be averted with fewer firearms in the US. We estimated that there would be approximately 26% fewer suicide fatalities, equivalent to 11,630 fewer suicide fatalities each year, if the US had firearms means restriction bringing ownership rates equal to those in Canada. Canada's main approach to restricting firearms is to require licenses for firearms possession. The licensing process requires individuals to have passed a firearm safety course and an additional restricted firearm safety course for firearms. The process also includes evaluation of suicide risk and risk of violence against others. An estimated 77% of the US public supports similar firearm licensing requirements in the US [35], suggesting that it would be feasible for US policymakers to pass such policies, and they would save more than 11,000 lives a year in the US [35]. Such an approach may be urgently called for, given a context of increasing US suicide fatalities over the past 17 years [36].

## Supporting information

**S1 Text. Detailed summary of methods.**
(DOCX)

**S1 Table. Canadian ethnicity distribution as estimated through the 2016 Canadian Census, among males aged 0–14.**
(DOCX)

**S2 Table. Firearm and non-firearm suicide deaths and denominators for the US in 2016 according to WISQARS among males aged 0–14, grouped into ethnicity categories to align with Canada.**
(DOCX)

**S3 Table. Calculated firearm and non-firearm suicide rates in the US, standardized to the ethnic distribution of Canada, 2016, males aged 0–14.**
(DOCX)

**S4 Table. All-cause suicide deaths in the US that would be averted if the US had the same suicide rates as in Canada, 2016, among males aged 0–14.**
(DOCX)

**S5 Table. Firearm and non-firearm suicide deaths, population, and crude rates in Canada, according to Statistics Canada, by age and sex, 2016.**
(DOCX)

**S6 Table. Firearm and non-firearm suicide deaths, population, and crude rates in the US, standardized to the ethnic distribution of Canada, by age and sex, 2016.**
(DOCX)

**S7 Table. Firearm and non-firearm suicide deaths, denominators, and crude rates for the US according to WISQARS, by age and sex, 2016, grouped into ethnicity categories to align those provided for the Canadian population.**
(DOCX)

## Author Contributions

**Conceptualization:** Julia Raifman, Sandro Galea.

**Formal analysis:** Julia Raifman, Laura Sampson.

**Methodology:** Julia Raifman, Laura Sampson.

**Writing – original draft:** Julia Raifman, Laura Sampson, Sandro Galea.

**Writing – review & editing:** Julia Raifman, Laura Sampson, Sandro Galea.

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
