## [Decision Letter · Decision Letter 0]

4 Feb 2020

PONE-D-19-27645

Suicide fatalities in the US compared to Canada: Potential suicides averted with lower firearm ownership in the US.

PLOS ONE

Dear Ms. Sampson,

Thank you for submitting your manuscript to PLOS ONE. After careful consideration, we feel that it has merit but does not fully meet PLOS ONE’s publication criteria as it currently stands. Therefore, we invite you to submit a revised version of the manuscript that addresses the points raised during the review process.

We would appreciate receiving your revised manuscript by Mar 19 2020 11:59PM. To enhance the reproducibility of your results, we recommend that if applicable you deposit your laboratory protocols in protocols.io, where a protocol can be assigned its own identifier (DOI) such that it can be cited independently in the future. For instructions see: http://journals.plos.org/plosone/s/submission-guidelines#loc-laboratory-protocols

We look forward to receiving your revised manuscript.

Kind regards,

Kyoung-Sae Na, M.D.

Academic Editor

PLOS ONE

Journal Requirements:

Reviewers' comments:

Reviewer's Responses to Questions

**Comments to the Author**

1. Is the manuscript technically sound, and do the data support the conclusions?

Reviewer #1: Yes

Reviewer #2: Partly

2. Has the statistical analysis been performed appropriately and rigorously? 

Reviewer #1: Yes

Reviewer #2: No

3. Have the authors made all data underlying the findings in their manuscript fully available?

Reviewer #1: Yes

Reviewer #2: Yes

4. Is the manuscript presented in an intelligible fashion and written in standard English?

Reviewer #1: Yes

Reviewer #2: Yes

5. Review Comments to the Author

Reviewer #1: This is a straightforward and methodologically sound paper. My only comment is that there is no discussion of a potential race-suicide effect measure modification by place. In the US, whites are significantly more likely than blacks to complete suicide for all methods of suicide. I would expect this to be consistent in Canada, but I am not sure; the magnitude of the association could also be different. I recognize that the current data prevent such an evaluation in Canada.

I do think it is important to consider, however, because the authors compare the US rates (race and sex standardized) to the Canadian suicide rates in order to estimate the number of suicides averted if the US had the same gun control practices as Canada. Effect measure modification by place may mean that the authors' estimates of potential suicides averted may be underestimated (for example if whites are more likely to commit suicide in Canada than in the US) or overestimated (for example if whites are less likely to commit suicide in Canada than in the US). If there are previous studies that evaluate the association between race and suicide in Canada, the authors may be able to determine if effect measure modification by place may have altered their results.

Overall, excellent work.

Reviewer #2: First, I would like to thank the authors for undertaking the effort to investigate the potential benefits of more restrictive firearm legislation within the US.

In the present paper, 'Suicide fatalities in the US compared to Canada: Potential suicides averted with lower firearm ownership in the US', the authors investigate a potential reduction in total suicide due to a potential reduction of firearm suicide rates in the US to the level of firearm suicide rates of Canada.

The publication is well written and the methods are described in detail. Furthermore, data are available as support material.

The topic itself is highly relevant and an important public health concern. I support publication of this article.

Some points should, however, be addressed:

- Throughout the article the authors refer to "race", "race and ethnicity" or "race/ethnicity". As the race homo sapiens sapiens only consist of one race with different ethnicities and the term is, therefore, false, I highly suggest deleting the term and only using "ethnicity". (even though the term is wrongly used in official documents in the US).

- When investigating the effect of change in firearm ownership on suicide rates: discussion and presentation of relevant data on Australia, Austria and Israel should be mentioned.

- As the US has no data on circulating firearms and/or firearm certificates, calculating an effect seems difficult. This should be explained in the limitations. Alternatively the authors may also use existing data on countries with a reduction in firearm availability and consecutive reduction in firearm suicides (see previous remark).

6. PLOS authors have the option to publish the peer review history of their article (what does this mean?). If published, this will include your full peer review and any attached files.

Reviewer #1: Yes: Aaron M Frutos

Reviewer #2: No

---

## [Author Response · Author response to Decision Letter 0]

31 Mar 2020

To: Kyoung-Sae Na, M.D.

Academic Editor

PLOS ONE

March 6, 2020

Dear Dr. Kyoung-Sae Na, 

We would like to thank you for the opportunity to revise our manuscript. The reviewer comments were extremely helpful, and we believe our paper is stronger now after implementing their suggestions. Below, we detail the changes we made and our responses to each point brought up by the reviewers. 

Reviewer #1: This is a straightforward and methodologically sound paper. My only comment is that there is no discussion of a potential race-suicide effect measure modification by place. In the US, whites are significantly more likely than blacks to complete suicide for all methods of suicide. I would expect this to be consistent in Canada, but I am not sure; the magnitude of the association could also be different. I recognize that the current data prevent such an evaluation in Canada.

I do think it is important to consider, however, because the authors compare the US rates (race and sex standardized) to the Canadian suicide rates in order to estimate the number of suicides averted if the US had the same gun control practices as Canada. Effect measure modification by place may mean that the authors' estimates of potential suicides averted may be underestimated (for example if whites are more likely to commit suicide in Canada than in the US) or overestimated (for example if whites are less likely to commit suicide in Canada than in the US). If there are previous studies that evaluate the association between race and suicide in Canada, the authors may be able to determine if effect measure modification by place may have altered their results.

Overall, excellent work.

Our response: Thank you for these comments and for pointing out the possibility of effect measure modification by place in the relationship between race (ethnicity) and suicide. Unfortunately, we are not aware of data on suicide death rates by black individuals compared to white individuals in Canada. Thus, we have added the following statement in the limitations section on page 16:

“Given that detailed suicide fatality data was not available by ethnicity in Canada, it is not clear whether the associations between ethnicity and suicide observed in the US (i.e., that white individuals are much more likely than black individuals to commit suicide) also hold in Canada. In other words, there may be effect modification by country in this relationship, which could affect our overall results and conclusions.”

Additionally, we edited a sentence in the introduction section on page 7 that previously stated, “Black populations typically have lower suicide fatality rates [25], and a greater proportion of the US compared to the Canadian population is Black” to now state, “Black populations typically have lower suicide fatality rates in the US [25], and a greater proportion of the US compared to the Canadian population is Black” (page 7), which is now more correct, since we do not know whether this statement is necessarily true in Canada in addition to the US. 

Reviewer #2: First, I would like to thank the authors for undertaking the effort to investigate the potential benefits of more restrictive firearm legislation within the US.

In the present paper, 'Suicide fatalities in the US compared to Canada: Potential suicides averted with lower firearm ownership in the US', the authors investigate a potential reduction in total suicide due to a potential reduction of firearm suicide rates in the US to the level of firearm suicide rates of Canada.

The publication is well written and the methods are described in detail. Furthermore, data are available as support material.

The topic itself is highly relevant and an important public health concern. I support publication of this article.

Our response: Thank you for these comments.

Some points should, however, be addressed:

1. Throughout the article the authors refer to "race", "race and ethnicity" or "race/ethnicity". As the race homo sapiens sapiens only consist of one race with different ethnicities and the term is, therefore, false, I highly suggest deleting the term and only using "ethnicity". (even though the term is wrongly used in official documents in the US).

Our response: Thank you for pointing this out. We agree, and we have deleted all references to “race” throughout the manuscript, tables, and appendix materials; we now only refer to ethnicity.

2. When investigating the effect of change in firearm ownership on suicide rates: discussion and presentation of relevant data on Australia, Austria and Israel should be mentioned.

Our response: Thank you for this suggestion. We have added the following text (along with new references) on page 5 of the introduction: 

“Other studies suggest there have been similar reductions in all-cause suicide fatalities after countries have changed federal policies to reduce firearm ownership or access. After Israel restricted weekend access to firearms for Israeli Defense Force soldiers, the suicide rate declined by 40% [11]. After Australia banned long guns and implemented a gun buyback program beginning in 1996, trends in all-cause suicide mortality reversed from increasing 1% percent per year to declining by 1.5% per year [12]. Finally, after Austria made handgun purchase policies more stringent in 1998, there was a decline in the suicide rate due to firearms, but not in the all-cause suicide rate [13].” 

Additionally, we added the following sentence on page 15 of the discussion section: 

“…The reduction in suicides associated with reduced firearms is also consistent with evidence of reduced all-cause suicides following federal policy changes in Israel and Australia [11,12].”

3. As the US has no data on circulating firearms and/or firearm certificates, calculating an effect seems difficult. This should be explained in the limitations. Alternatively the authors may also use existing data on countries with a reduction in firearm availability and consecutive reduction in firearm suicides (see previous remark).

Our response: We agree with this limitation in our analyses, and we should have been more clear in stating this in the original version. We have added the following sentence to page 16-17in the limitations section:

“Finally, the US does not have data on circulating firearms or firearm certificates, and ultimately we cannot measure the exact effect of a change in firearm availability or ownership on suicide deaths. Our conclusions rest on the assumption that if the US had similar firearm ownership rates as in Canada, the firearm suicide death rates in the US would be comparable to those in Canada, given the many other similarities between the two countries.”

Note: We have also made the required formatting changes listed in the email from the editor. Finally, we have changed the corresponding author to be the first author, Dr. Julia Raifman. Ms. Sampson was made corresponding author for the initial submission only, due to Dr. Raifman’s maternity leave at the time. 

Thank you again to each of the reviewers and the editor.

Sincerely, 

Julia Raifman, ScD

Boston University School of Public Health

Boston, Massachusetts

---

## [Decision Letter · Decision Letter 1]

13 Apr 2020

Suicide fatalities in the US compared to Canada: Potential suicides averted with lower firearm ownership in the US.

PONE-D-19-27645R1

Dear Dr. Raifman,

We are pleased to inform you that your manuscript has been judged scientifically suitable for publication and will be formally accepted for publication once it complies with all outstanding technical requirements.

With kind regards,

Kyoung-Sae Na, M.D.

Academic Editor

PLOS ONE

**Comments to the Author**

1. If the authors have adequately addressed your comments raised in a previous round of review and you feel that this manuscript is now acceptable for publication, you may indicate that here to bypass the “Comments to the Author” section, enter your conflict of interest statement in the “Confidential to Editor” section, and submit your "Accept" recommendation.

Reviewer #2: All comments have been addressed

2. Is the manuscript technically sound, and do the data support the conclusions?

Reviewer #2: Yes

3. Has the statistical analysis been performed appropriately and rigorously? 

Reviewer #2: Yes

4. Have the authors made all data underlying the findings in their manuscript fully available?

Reviewer #2: Yes

5. Is the manuscript presented in an intelligible fashion and written in standard English?

Reviewer #2: Yes

6. Review Comments to the Author

Reviewer #2: Regarding the Revision of "Suicide fatalities in the US compared to Canada: Potential suicides averted with lower firearm ownership in the US." I reiterate, that the publication is well written and the topic of importance.

I had only later remembered, that also data on Switzerland (where legislation changes similar to the reform in Israel was passed) may be of interest or that newer data on Austria (more in line with the position of the authors results would also be available).

While the addition of these works to the discussion may of course be of interest to the authors, it does not warrant a mandatory revision in my opinion.

I advise to accept the manuscript in its revised form and thank the authors for a valuable contribution.

7. PLOS authors have the option to publish the peer review history of their article (what does this mean?). If published, this will include your full peer review and any attached files.

Reviewer #2: No

---

## [Editor Report · Acceptance letter]

20 Apr 2020

PONE-D-19-27645R1 

Suicide fatalities in the US compared to Canada: Potential suicides averted with lower firearm ownership in the US 

Dear Dr. Raifman:

I am pleased to inform you that your manuscript has been deemed suitable for publication in PLOS ONE. Congratulations! Your manuscript is now with our production department. 

With kind regards,

on behalf of

Dr. Kyoung-Sae Na 

Academic Editor

PLOS ONE